# Microbiome of Saliva and Plaque in Children According to Age and Dental Caries Experience

**DOI:** 10.3390/diagnostics11081324

**Published:** 2021-07-23

**Authors:** Eungyung Lee, Suhyun Park, Sunwoo Um, Seunghoon Kim, Jaewoong Lee, Jinho Jang, Hyoung-oh Jeong, Jonghyun Shin, Jihoon Kang, Semin Lee, Taesung Jeong

**Affiliations:** 1Department of Pediatric Dentistry, Dental Research Institute, Pusan National University Dental Hospital, Yangsan 50612, Korea; leeungyung@gmail.com (E.L.); jonghyuns@pusan.ac.kr (J.S.); 2Department of Biomedical Engineering, College of Information-Bio Convergence Engineering, Ulsan National Institute of Science and Technology (UNIST), Ulsan 44919, Korea; suhyunpark@unist.ac.kr (S.P.); hoonkim@unist.ac.kr (S.K.); jwlee230@unist.ac.kr (J.L.); jjh2379@unist.ac.kr (J.J.); hyoung-oh@unist.ac.kr (H.-o.J.); 3Helixco Inc., Ulsan 44919, Korea; swum@helixco.co.kr; 4Department of Pediatric Dentistry, School of Dentistry, Institute of Translational Dental Science, Pusan National University, Yangsan 50612, Korea

**Keywords:** children, dental caries, microbiome, next-generation sequencing

## Abstract

Dental caries are one of the chronic diseases caused by organic acids made from oral microbes. However, there was a lack of knowledge about the oral microbiome of Korean children. The aim of this study was to analyze the metagenome data of the oral microbiome obtained from Korean children and to discover bacteria highly related to dental caries with machine learning models. Saliva and plaque samples from 120 Korean children aged below 12 years were collected. Bacterial composition was identified using Illumina HiSeq sequencing based on the V3–V4 hypervariable region of the 16S rRNA gene. Ten major genera accounted for approximately 70% of the samples on average, including *Streptococcus*, *Neisseria*, *Corynebacterium*, and *Fusobacterium*. Differential abundant analyses revealed that *Scardovia wiggsiae* and *Leptotrichia wadei* were enriched in the caries samples, while *Neisseria oralis* was abundant in the non-caries samples of children aged below 6 years. The caries and non-caries samples of children aged 6–12 years were enriched in *Streptococcus mutans* and *Corynebacterium durum*, respectively. The machine learning models based on these differentially enriched taxa showed accuracies of up to 83%. These results confirmed significant alterations in the oral microbiome according to dental caries and age, and these differences can be used as diagnostic biomarkers.

## 1. Introduction

Dental caries is a common infectious disease that affects people of all ages and both genders worldwide. Although the prevalence of dental caries is decreasing, it can still be found in more than 50% of children [1,2]. The mechanism of dental caries fundamentally comes from the balance of demineralization and remineralization in the tooth structures [3]. When the pH in the oral cavity is lowered below a certain level due to the acid produced by the microbes in the plaque, demineralization occurs in which the mineral components in the hard tissue are dissolved [3]. Conversely, when the oral pH is restored to the neutral level, dissolved calcium and phosphorus components are deposited again. This process is called remineralization [3].

*Streptococcus mutans*, *Streptococcus sobrinus*, and *Lactobacillus* are the representative bacteria known to be associated with dental caries [4]. However, only about 50% of approximately 700 types of oral microorganisms have been cultivated and named. It is also known that the predominant microbes causing dental caries differ by race and age, and even differ among people of the same race [5,6,7,8].

The microbiome refers to the sum of all microorganisms and their genomes in a particular environment [9]. Oral cavity is the second largest microbial community in the humans after the gut; a significant relationship has been found between the microbes in the oral cavity and the onset of some systemic diseases [5,10,11]. Therefore, understanding the oral microbiome will make it possible to understand not only oral infectious diseases, such as dental caries and periodontal diseases, but also cardiovascular diseases, stroke, rheumatoid arthritis, cancer, chronic obstructive pulmonary disease, and metabolic diseases such as diabetes and obesity [5,10,11].

Microbiome research has developed with the advances in molecular biology technology and next-generation sequencing (NGS). In the field of dentistry, NGS has been used to analyze the oral microbes involved in infectious diseases, including dental caries, and to identify the genes associated with oral diseases such as hypodontia and oral cancer [12,13,14,15,16]. Additionally, longitudinal changes in the oral microbiome of children from 3 months to 7 years after birth were confirmed by NGS [17,18,19]. The distribution of site-specific oral microbiomes in the oral cavity has also been discovered [20,21]. Although it is widely known that oral microbes are one of the main determinants of dental caries [22], there are only few reports on the oral microbiome of Korean children, especially of children and adolescents under the age of 12 years, who are the primary patients in pediatric dentistry [4,23,24,25,26,27].

Recently, machine learning techniques have been actively applied to analyzing oral microbiota for making predictions of health status based on the hidden patterns in a large amount of metagenomics data. For example, Omori et al. developed random forest models to classify type 2 diabetes mellitus and a healthy control based on metagenomic profiles of saliva [28]. Kato-Kogoe et al. performed 16S rRNA metagenomic analysis of salivary microbiota in patients with atherosclerotic cardiovascular disease and proposed random forest models for predicting atherosclerotic cardiovascular disease [29]. Although machine learning approaches have also been adopted for predicting childhood caries based on oral microbiome in several previous studies, they were often based on a limited number of samples and focused on a certain ethnic group [30,31,32].

The aim of this study was to analyze the oral microbiome of saliva and plaque collected from 120 Korean children under 12 years of age. The children were categorized into non-caries and caries groups according to their dental caries experience. The richness and diversity of the bacterial community were analyzed using NGS. Moreover, machine learning models were developed to classify the non-caries and caries samples.

## 2. Materials and Methods

### 2.1. Patient Selection and Oral Examination

The participants were 120 children who visited the Department of Pediatric Dentistry, Pusan National University Dental Hospital, Yangsan, Korea. Group 1 included children aged below 6 years, and Group 2 included children aged 6–12 years in accordance with oral health surveys of the World Health Organization (WHO) [33]. The number of samples was determined with G*power software (ver.3.1.9, Düsseldorf, Germany) to maintain a significance level of 0.05% and power of 0.8 based on previous studies [34,35]. The minimum samples were calculated as 24 for each group; 30 subjects participated in this study considering the dropout rate of 10% or more. Inclusion criteria were that children were medically healthy, had at least completed primary dentition with the primary second molars, and had not used antibiotics within the preceding 2 weeks. There was no age limit, but children with a lack of compliance for clinical examination and sample collection were excluded. Children without complete primary dentition were excluded. Furthermore, children with systemic diseases or disabilities that influence oral health care ability, salivary gland dysfunction, and fixed orthodontic devices which can affect alterations of the oral microbiome were excluded [36].

All clinical examinations were performed by a pediatric dentist with the subject lying in the supine position on a dental chair. The presence of dental caries was recorded using the International Caries Detection and Assessment System (ICDAS) [20,37]. Patients whose teeth were graded as ICDAS code 0–2 only were classified as the non-caries group, and those with ICDAS code 3 or higher as the caries group.

Written informed consent was obtained from all participants involved in the study. The study design, protocol, and informed consent were approved by the Institutional Review Board of Pusan National University Dental Hospital (PNUDH-2018-024, 22 August 2018).

### 2.2. Sample Preparation and DNA Extraction

All participants were instructed not to clean their teeth the evening and morning before sampling and not to eat or drink for 2 h preceding the sampling in the morning [25,27,38]. Supragingival plaque specimens were scraped using a sterile Gracey curette and pooled into sterile Eppendorf tubes. Plaque samples were primarily collected on primary second molars in children below 6 years, and permanent first molars in children aged 6–12 years old. Saliva samples were collected by rinsing the mouth for 30 s with 12 mL of a solution (E-zen Gargle, JN Pharm, Korea) [39]. All procedures related to the sample collection were performed in a safe environment under the supervision of the principal investigator. The plaque samples were immediately stored at −20 °C and saliva samples at 4 °C until further processing. In total, 240 samples were prepared for subsequent processing (Figure 1).

ICDAS: International Caries Detection and Assessment System; DMFT: Decayed, Missing, Filled Teeth index by the World Health Organization criteria; N1P, plaque sample of the non-caries group of Group 1; N1S, saliva sample of the non-caries group of Group 1; N2P, plaque sample of the non-caries group of Group 2; N2S, saliva sample of the non-caries group of Group 2; C1P, plaque sample of the caries group of Group 1; C1S, saliva sample of the caries group of Group 1; C2P, plaque sample of the caries group of Group 2; C2S, saliva sample of the caries group of Group 2.

For DNA extraction of the saliva, 8 mL of the gargled solution was transferred to a 15 mL conical tube and centrifuged at 3900 rpm for 10 min to obtain the precipitate. The supernatant was discarded and 200 μL of phosphate-buffered saline was added to the precipitate to obtain a resuspended sample. The completely resuspended sample was transferred to a 1.5 mL microcentrifuge tube, and DNA was extracted using an Exgene Clinic SV DNA extraction kit (GeneAll^®^, Seoul, Korea) according to the manufacturer’s instructions. The mixture of 20 μL proteinase K solution and 200 μL of BL buffer was added to the resuspended sample and vortexed vigorously. The sample was incubated at 56 °C for 10 min, then spun down briefly to remove any drops form inside of the lid. We added 200 μL of absolute ethanol to the sample, vortexed and spun down in the same way. The mixture was carefully transferred to the SV column and centrifuged for 10 min at 13,500 rpm, then replaced the collection tube with a new one. The sample was added 600 μL of buffer BW, centrifuged for 1 min at the same speed and replaced the collection tube with a new one. We applied 700 μL of buffer TW and centrifuged the sample for 1 min. Then we discarded the solution in the collection tube and reinserted the SV column back into the collection tube. The sample was centrifuged for 1 min to remove residual buffer and placed the SV column in a new 1.5 mL microcentrifuge tube and 50 μL elution buffer was added. The sample was incubated at room temperature for 1 min and then centrifuged for 1 min for the elution of DNA.

For DNA extraction of the plaque sample, we melt the sample at room temperature and agitated to lyse the sample after adding 200 μL of buffer CL. We added 20 μL of proteinase K in the tube and then vortexed the mixture and spun down to remove the bubbles inside the tube. The sample was incubated overnight at 56 °C and transferred the solution inside the tube to a new one. We added 200 μL of buffer BL and vortexed, rotated for a while to remove the bubbles inside the lid. After adding 200 μL of absolute ethanol and transferring it to the SV column, DNA was extracted using the same method as the saliva samples.

The DNA quality and quantity were determined using NanoDrop spectrophotometer (Thermo Fisher Scientific, Waltham, MA, USA).

### 2.3. Polymerase Chain Reaction Amplification of 16S rRNA Genes and Sequencing Data Analysis

Polymerase chain reaction (PCR) amplification was performed using primers targeting regions of the hypervariable regions V3 and V4 of the 16S rRNA genes. The 341F (5′-CCTACGGGNGGCWGCAG-3′) and 806R (5′-GACTACHVGGGTATCTAATCC-3′) primers were designed and used. The amplified product was quantified according to the quantitative PCR Quantification Protocol Guide (KAPA Library Quantification kits for Illumina Sequencing platforms) and was verified using HT DNA High Sensitivity LabChip^®^ GX Kit (Caliper, PerkinElmer, Hopkinton, MA, USA). Thereafter, paired-end (2 × 250 bp) sequencing was performed using the HiSeq™ platform (Illumina, San Diego, CA, USA). Sequencing data were analyzed using QIIME2, version 2020.6 [40]. The sequences were denoised and clustered without trimming using Deblur [40], which uses a novel sub-operational-taxonomic-unit approach. In the diversity analysis, rarefaction was needed to normalize the difference in the frequency among samples, which randomly subsampled the same number of sequences from each sample. The alpha diversity indices of Shannon index, Faith’s phylogenetic diversity (PD), Observed features, and Pielou’s evenness were calculated using QIIME2. The Jaccard distance was used in the principal coordinate analysis (PCoA) plot to explain dissimilarity, which was drawn using the QIIME viewer [40]. The taxonomy was determined at level 6 (genus) and level 7 (species) using the SILVA 16S rRNA database and the *qiime taxa collapse* method, such that each sample had assigned bacteria according to their sequence. The frequency of assigned bacteria was divided by the total bacterial frequency in each sample (i.e., the percentage of the assigned bacteria in each sample). A heatmap showing the major 10 genera was drawn using R software package, version 3.6.1 (available online: https://www.r-project.org (accessed on 18 June 2021).

### 2.4. Statistical Analysis

An independent *t*-test was used after the normality test with the Shapiro–Wilk test to compare the differences in age, gender, and dental caries indices using Statistical Product and Service Solutions, version 22.0 (IBM SPSS, Armonk, NY, USA). Alpha diversity indices of samples were compared using Student’s *t*-test. The Wilcoxon rank-sum test was performed to compare the differences in the microbiomes according to age and dental caries experience from the phylum level to the species level. Based on the results of taxonomic assignment, data showing significant differences between the groups were selected using the Kruskal–Wallis test, and linear discriminant analysis (LDA) for the selected data was conducted. The LDA effect size (LEfSe) method was used to analyze the differences in the microbial community composition between the non-caries and caries groups. The LDA cutoff score was set at 2.0, and *p*-values <0.05 were considered statistically significant.

### 2.5. Machine Learning Models for Classifying Non-Caries and Caries Samples

The machine learning models used in this study followed the default option. Random forest was used to classify the non-caries and caries samples in each group. Bacteria at the genus and species levels from differential abundance analysis were used as features in the models. Each feature was evaluated based on its importance in each group using random forest models (Appendix A). The features were added one by one in order of importance from the highest to the lowest, resulting in many models with various feature combinations (Appendix A), as previously done by Kim et al. [41]. The models with the best accuracy were selected. The models performed stratified 5-fold cross-validation, which divided the data into five subsets with one subset for the test set and four for the training set. The performances of the models were evaluated by calculating the confusion matrix of accuracy, balanced accuracy, precision, sensitivity, specificity, and standard deviations. The scope of the confusion matrix ranged between 0 and 1.
$$\mathrm{Accuracy}=\frac{\mathrm{TP}+\mathrm{TN}}{\mathrm{TP}+\mathrm{TN}+\mathrm{FP}+\mathrm{FN}}\;\mathrm{Balanced}\;\mathrm{accuracy}=\frac{\mathrm{TP}}{2\times \left(\mathrm{TP}+\mathrm{FN}\right)}+\frac{\mathrm{TN}}{2\times \left(\mathrm{TN}+\mathrm{FP}\right)}\;\mathrm{Precision}=\frac{\mathrm{TP}}{\mathrm{TP}+\mathrm{FP}}\;\mathrm{Sensitivity}=\frac{\mathrm{TP}}{\mathrm{TP}+\mathrm{FN}}\;\mathrm{Specificity}=\frac{\mathrm{TN}}{\mathrm{TN}+\mathrm{FP}}$$
where TP is true positive; TN, true negative; FP, false positive; and FN, false negative.

## 3. Results

This study enrolled 120 Korean children. The mean age in Group 1 (25 boys and 35 girls aged below 6 years) was 4.5 ± 0.7 (age range 3.1–5.9) years, and that in Group 2 (35 boys and 25 girls aged 6–12 years) was 7.9 ± 1.3 (age range 6.0–11.8) years. There was no significant difference in the mean age according to gender between the groups (Table 1). The mean ICDAS score of the non-caries group and caries group was 0.30 ± 0.71 and 4.12 ± 0.7, respectively, and the caries index between the two groups differed significantly (*p* < 0.05) (Table 2).

### 3.1. Alpha and Beta Diversity Analysis

There were significant differences between Group 1 and Group 2 in the alpha diversity indices of Shannon index, observed features, Faith’s PD, and Pielou’s evenness (Figure 2). The bacterial communities were more diverse and evenly distributed in Group 2 than in Group 1 (Figure 2a,d). Furthermore, the number of bacteria in Group 2 was higher than that in Group 1 (Figure 2b). Faith’s PD was higher in Group 2 than in Group 1, indicating a greater biodiversity in the former than in the latter (Figure 2c).

A PCoA plot based on Jaccard distance metrics showed that the samples were clustered by plaque or saliva, indicating dissimilarity in the bacterial compositions between the plaque and saliva samples (Figure 3).

### 3.2. Bacterial Compositions

A total of 11 phyla, 19 classes, 34 orders, 68 families, 143 genera, and 377 species were detected in the Group 1 samples. At the phylum level, the five most abundant phyla were Firmicutes, Proteobacteria, Actinobacteria, Bacteroidetes, and Fusobacteria in both non-caries and caries groups (Figure 4).

Group 1 included children aged below 6 years.

A total of 14 phyla, 21 classes, 37 orders, 72 families, 154 genera, and 421 species were detected in the Group 2 samples. At the phylum level, the five most abundant phyla were Firmicutes, Proteobacteria, Actinobacteria, Bacteroidetes, and Fusobacteria in both the non-caries and caries groups, similar to those in Group 1 (Figure 5).

Group 2 included children aged from 6 to 12 years.

The dominant 10 genera are shown in Figure 6. *Streptococcus* (20.5%), *Neisseria* (8.2%), *Corynebacterium* (7.0%), *Leptotrichia* (6.5%), *Actinomyces* (6.4%), *Capnocytophaga* (5.2%), *Rothia* (4.9%), *Haemophilus* (4.3%), *Prevotella* (4.0%), and *Fusobacterium* (3.5%) accounted for approximately 70.5% of the bacteria in each sample on an average. *Streptococcus* predominated in all samples, and its proportion was higher in the saliva samples than in the plaque samples.

### 3.3. Differential Abundance Analysis

To identify the bacteria that were differentially abundant in each non-caries group and caries group, LEfSe analysis was performed between the two groups considering the plaque or saliva samples.

Figure 7a shows the differential abundance of the bacteria in the plaque samples of the non-caries and caries groups of Group 1 (N1P and C1P, respectively). At the species level, *Corynebacterium matruchotii*, *Capnocytophaga granulosa*, *Leptotrichia wadei*, *Prevotella salivae*, *Anaeroglobus geminatus*, *Selenomonas sputigena*, *Scardovia wiggsiae*, and *Prevotella histicola* were abundant in the C1P group. At the genus level, *Leptotrichia*, *Veillonella*, *F0332 uncultured bacterium*, *Lachnoanaerobaculum*, *Gemella uncultured organism*, *JGI 0000069 P22*, *Bergeyella*, and *F0332* were abundant in the C1P group. At the species level, *Neisseria oralis* was abundant in the N1P group.

Figure 7b shows the differential abundance of bacteria in the plaque samples of the non-caries and caries groups of Group 2 (N2P and C2P, respectively). At the species level, *Treponema maltophilum*, *Dialister pneumosintes*, *Capnocytophaga haemolytica*, *Anaeroglobus geminatus*, *Candidatus saccharibacteria*, *Streptococcus mutans*, and *Porphyromonas pasteri* were abundant in the C2P group. At the genus level, *Porphyromonas*, *Lachnoanaerobaculum*, *F0058 uncultured bacterium*, *uncultured Peptococcus*, *Clostridia UCG 014 Clostridiales bacterium*, *Centipeda uncultured bacterium*, *Candidatus Saccharimonas*, *Lactobacillus*, *Bergeyella*, *Treponema*, *Peptostreptococcus*, and *Granulicatella* were abundant in the C2P group. At the species level, *Corynebacterium durum*, *Aggregatibacter* sp., and *Treponema socranskii* were abundant in the N2P group. At the genus level, *Actinomyces*, *Kingella*, *Cardiobacterium uncultured bacterium*, and *Kingella uncultured bacterium* were abundant in the N2P group.

Figure 8a shows the differential abundance of bacteria in the saliva samples of the non-caries and caries groups of Group 1 (N1S and C1S, respectively). At the species level, *Corynebacterium matruchotii*, *Campylobacter gracilis*, *Prevotella oulorum*, *Lactobacillus fermentum*, *Capnocytophaga granulosa*, *Selenomonas flueggei*, *Selenomonas noxia*, *Leptotrichia wadei*, *Scardovia wiggsiae*, and *Selenomonas sputigena* were abundant in the C1S group. At the genus level, *Aggregatibacter*, *F0332 uncultured bacterium*, *Shuttleworthia*, *Tannerella*, *uncultured Atopobium*, *Granulicatella*, *F0332*, and *JGI 0000069 P22* were abundant in the C1S group. At the species level, *Neisseria oralis*, *Veillonella atypica*, and *Haemophilus quentini* were abundant in the N1S group. At the genus level, *Streptococcus*, *Enhydrobacter*, and *Pseudopropionibacterium uncultured bacterium* were abundant in the N1S group.

Figure 8b shows the differential abundance of bacteria in the saliva samples of the non-caries and caries groups of Group 2 (N2S and C2S, respectively). At the species level, *Streptococcus mutans*, *Prevotella pallens*, *Prevotella veroralis*, *Anaeroglobus geminatus*, *Lactobacillus fermentum*, *Megasphaera micronuciformis*, *Prevotella buccae*, *Alloprevotella rava*, *Parascardovia denticolens*, *Olsenella uli*, and *Streptococcus sobrinus* were abundant in the C2S group. At the genus level, *Lactobacillus*, *Bifidobacterium*, *Selenomonas Veillonellaceae bacterium*, *Atopobium*, *Absconditabacteriales SR1*, *Dialister*, *Staphylococcus*, and *Bergeyella* were abundant in the C2S group. At the species level, *Corynebacterium durum*, *Actinomyces massiliensis*, *Capnocytophaga gingivalis*, and *Cardiobacterium hominis* were abundant in the N2S group. At the genus level, *Actinomyces*, *Lautropia uncultured bacterium*, *Kingella*, *Cardiobacterium uncultured bacterium*, *Moraxella*, and *Stenotrophomonas* were abundant in the N2S group.

### 3.4. Classification of Samples Using the Random Forest Machine Learning Model

Taxa at the genus or species level in each group from the differential abundance analysis were used as features that classified the non-caries and dental caries samples in the random forest machine learning models. The selected taxa were evaluated based on their importance in each group using the random forest model (Appendix A). The taxa were then added one by one in the order of importance from the highest to the lowest in the machine learning model to identify the best feature combination that resulted in the highest accuracy (Appendix A). The best feature (taxa) combination, accuracy, balanced accuracy, precision, sensitivity, and specificity in each group are shown in Table 3. The models showed accuracy over 0.7 in all cases, especially in the case of the saliva samples of Group 2 (accuracy of 0.83).

*Cardiobacterium uncultured bacterium* was used as one of the features in N2P vs. C2P and N2S vs. C2S, and 36 feature ids designated this bacterium in our data. One of the 36 feature ids showed the highest frequency (60,446), and the National Center for Biotechnology Information (NCBI) assigned it as Cardiobacterium hominis.

*Kingella uncultured bacterium* was used as one of the features in N2P vs. C2P, and 10 feature ids designated this bacterium in our data. One of the 10 feature ids showed the highest frequency (43,193), and NCBI assigned it as Kingella oralis.

## 4. Discussion

Oral bacteria in dental plaque biofilms are essential for the initiation and progression of dental caries [42,43]. Saliva is considered a suitable sample for studying oral microorganisms, because the sample is easy to collect and contains various microorganisms [38]. This study was performed to verify the differences in the abundance and diversity of microbes in the dental plaque and saliva in children.

Traditionally, the diagnosis of dental caries is made using visual, tactile, and radiographic methods. Recently, laser fluorescence or quantitative optical fluorescence has also been used [44,45]. ICDAS was initially proposed to monitor the progression of dental caries more accurately, manage dental caries more effectively, and facilitate communication between the clinicians and patients; ICDAS-II is currently being used [46,47]. Braga et al. analyzed the correlation between the WHO criteria and ICDAS-II and reported that ICDAS-II is feasible in epidemiological surveys in preschool children [48]. In this study, all teeth were graded according to ICDAS, and the decayed teeth were determined using the WHO criteria. Although there is a limitation that the evaluation depends solely on the knowledge and assessment of a trained examiner, it is a useful method for diagnosing dental caries in situations where radiological or histological examination is not possible [48,49].

Group 2 showed more richness and evenness as compared to Group 1 in the alpha diversity analysis. This could be due to the presence of permanent teeth in the former, which are less reactive to acid [21,50]. Previous studies have reported that dental plaque and saliva have different microbial compositions [51,52,53]. Our data also showed dissimilarity in the microbial compositions between the plaque and saliva samples.

Firmicutes was commonly detected as the most dominant phylum in all groups, which is consistent with the findings of previous studies [18,38,54,55]. Xu et al. reported that Proteobacteria was the most predominant in primary dentition and adults with permanent dentition, while Firmicutes was abundant in young adults with permanent dentition [21]. Chen et al. also reported a decrease in Firmicutes and an increase in Proteobacteria in the children’s group and an increase in Firmicutes and a decrease in Proteobacteria in the youth and adult groups [17]. In this study, the relative proportions of Firmicutes, Proteobacteria, and Bacteroidetes were similar in both groups. Research on changes in the microbial community in humans, according to age from adolescence to adulthood, is needed in the near future.

The Fusobacteria phylum was significant only in the plaque samples of Group 1 by LDA. Fusobacteria are classified as obligate anaerobes that are frequently observed in the gingival sulcus and deep gingival pockets [56,57]. Changes in the oral environment with increasing age can affect the oral flora. The eruption of the primary tooth is regarded as a major change, because the formation of the gingival sulcus leads to site-specific bacterial composition in addition to the presence of hard tissue in the oral cavity [58]. Shi et al. reported that Firmicutes were more abundant in saliva, while Fusobacteria and Actinobacteria were more abundant in plaque [57]. Shi et al. reported that permanent tooth sites tend to host more diverse bacterial communities as compared to the primary tooth sites [59]. They also found that Proteobacteria, Firmicutes, Bacteroidetes, Fusobacteria, and Actinobacteria varied highly among the sites [59]. The location and shape of the teeth in the oral cavity can affect the distribution of the anaerobic bacteria such as Fusobacteria. In children with mixed dentition, the time of co-existence of the deciduous and permanent teeth in the oral cavity can also be a factor affecting the microbial composition and ratio. In this study, the plaque samples from Group 2 were mainly collected from the permanent first molars. The different times of attachment and maturation of the bacteria to the teeth in the mixed dentition could be one of the reasons for the differences in the microbial composition and diversity [59].

*Streptococcus mutans*, a common caries pathogen, was enriched in the C2S and C2P groups, while another known oral pathogen, *Streptococcus sobrinus*, was enriched in the C2S group. Contrary to our results, a study of oral microbiome from adolescents aged 15 to 17 years reported that *Streptococcus mutans* and *Streptococcus sobrinus* had no significant associations with carious risk [60]. In addition, another study of the oral microbiome detected *Streptococcus mutans* at 28% of interdental biofilms from caries-free adults (20–35 years old) [61]. *Anaeroglobus geminatus* was enriched in the C1P, C2P, and C2S groups. Since this bacterium appeared in both Group 1 and Group 2, and it has previously been reported to be associated with oral disease [62], it could be a potential dental caries biomarker.

Badet et al. reported that *Lactobacillus* was the main pathogen for caries development [63], and Caufield et al. showed that *Lactobacillus fermentum* was dominant in adult and childhood caries [64]. Similarly, in our results, Lactobacillus was enriched in the C2P and C2S groups, and Lactobacillus *fermentum* was enriched in the C1S and C2S groups.

Idate et al. found that *Capnocytophaga granulosa* and *Capnocytophaga gingivalis* were more prevalent in healthy individuals than in patients with periodontitis [65]. In contrast, our data showed that *Capnocytophaga granulosa* was enriched in the C1P and C1S groups and *Capnocytophaga gingivalis* was enriched in the N2S group.

*Corynebacterium matruchotii* has been reported as enriched in the naturally healthy adult group aged from 22 to 67 years using LEfSe [66]. Qudeimat et al. found that *Corynebacterium matruchotii* and *Corynebacterium durum* were abundant in a caries-free group of 7-year-old children [67]. However, in our study, *Corynebacterium matruchotii* was enriched in the C1P and C1S groups, which was in contrast to the findings of previous reports. However, *Corynebacterium durum* was enriched in the N2P and N2S groups, which was consistent with previous reports.

*Neisseria oralis*, which is classified into the Proteobacteria phylum, was abundant in both plaque and saliva samples of the non-caries group of Group 1. Cherkasov et al. reported that *Neisseria oralis* was significantly enriched in the caries-depleted children’s group [13]. Another study showed that *Neisseria oralis* was also isolated from the gingival plaque of seven healthy people aged 1–80 years [68]. Furthermore, *Neisseria oralis* was used as one of the features in the machine learning models in this study, which classified the saliva samples from the caries group of Group 1 (Table 3). Abundance of *Neisseria oralis* could indicate it to be one of the bacteria that can maintain a healthy oral microbiome.

*Kingella uncultured bacterium* was assigned as *Kingella oralis* by NCBI, which was used as one of the features in N2P vs. C2P. Cherkasov et al. studied the microbiota of asthmatic children with or without dental caries and showed that *Kingella oralis* was one of the statistically significant caries-enriched species [13].

Age can play an important role in various oral diseases [69]. For example, clinicians can easily detect some oral lesions such as burning mouth syndrome and chronic candidiasis in the elderly rather than in young population [69]. The elderly who have decreased salivary flow rate are susceptible to some oral diseases because saliva acts as an innate immune system through its antimicrobial effect [70]. Some systemic conditions and diseases can bring about delayed wound healing in the oral cavity after surgical dental intervention as well for this reason [70]. Saliva has a great potential in remineralization of the tooth surface because it contains calcium and phosphate in concentrations [71]. Solomon et al. observed that higher values of salivary calcium in those with few carious lesions [71]. There is a common consensus that some systemic diseases such as rheumatoid arthritis, diabetes, and cardiovascular diseases are extremely closely related to *Porphyromonas gingivalis*, the well-known periodontal pathogens [11,72]. Patients with periodontitis and diabetes simultaneously are prone to developing complications of cardiovascular, renal, and retinopathic diseases [72]. Therefore, it is important to control the oral health and to maintain the balance of ecosystem between hosts and microbiota. In this respect, microbiome of saliva and plaque can be used as biomarkers for dental caries and periodontitis.

The main limitations of this study can be attributed to its cross-sectional design. We found differences in the composition and diversity of the microbial communities between the non-caries and caries groups. However, these findings do not indicate any causal relationships with respect to dental caries. Furthermore, we did not take into account socioeconomic status or the habits of oral hygiene or diet. These are known as important risk factors of dental caries [22]. Dietary habits are possible contributors of ethnic or racial variations in microbiome as well [73]. Another limitation is that only the plaque samples collected from dental hard tissue were used for analysis, although the distribution of microorganisms varies depending on the oral cavity. Particularly, children in the non-caries group had good oral hygiene; hence, plaque samples had to be collected inevitably from unspecified teeth, although they harbored a small amount of plaque. Therefore, it is necessary to consider sample collection based on the microbial community of the oral cavity. Future studies should analyze the changes in the oral flora before and after treatment of carious teeth and explore the longitudinal changes in the oral flora according to the growth and age of children.

## 5. Conclusions

It was confirmed that the composition and diversity of the oral microbiome differed significantly not only according to the types of samples collected, i.e., saliva and plaque, but also according to age. These results provide a theoretical basis for preventing and managing dental caries in pediatric dentistry and emphasize the use of microbial examination as a tool for diagnosing dental caries in clinics.

## Figures and Tables

**Figure 1 diagnostics-11-01324-f001:**
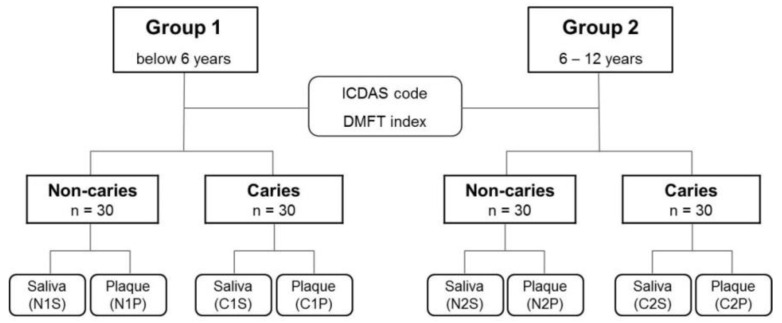
Group categorization and sample preparation flowchart. Group 1 included children aged below 6 years, and Group 2 included children aged 6–12 years. Each group was further divided into non-caries and caries subgroups according to the ICDAS code and DMFT index. Saliva and plaque samples were obtained from all participants. The sample coding is noted in each schematic box.

**Figure 2 diagnostics-11-01324-f002:**
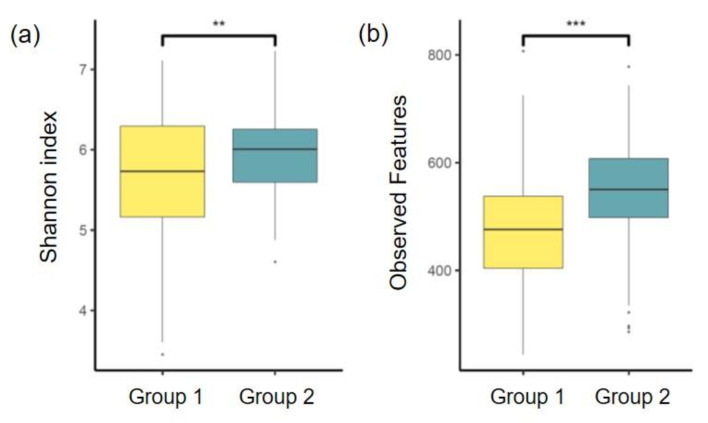
Comparison of the alpha diversity indices between Group 1 and Group 2 using student’s Table 1. and Group 2. (**a**) Distribution of Shannon index in group 1 and group 2; (**b**) Distribution of observed features in group 1 and group 2; (**c**) Distribution of Faith’s PD in group 1 and group 2; (**d**) Distribution of Pielou’s evenness in group 1 and group 2; * *p* < 0.05, ** *p* < 0.01, *** *p* < 0.001. Faith’s PD, Faith’s phylogenetic diversity.

**Figure 3 diagnostics-11-01324-f003:**
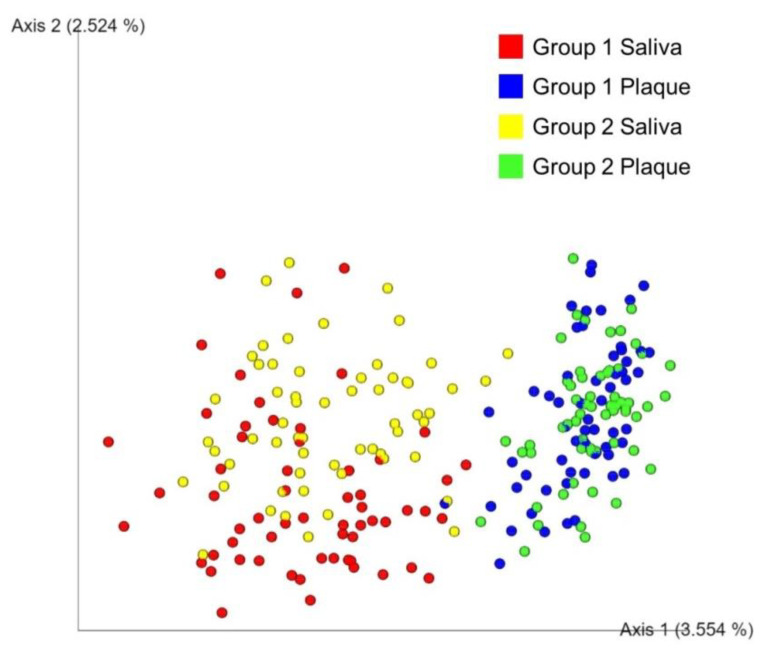
A principal coordinate analysis plot based on the Jaccard distance metrics showing dissimilarity between plaque and saliva samples of group 1 and group 2. The dots represent each plaque or saliva sample of group 1 or group 2.

**Figure 4 diagnostics-11-01324-f004:**
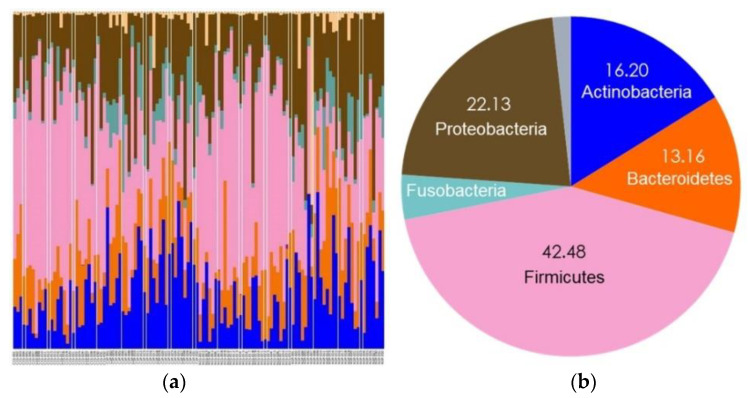
Taxonomic assignment at the phylum level in Group 1. (**a**) All taxa from the 120 saliva and plaque samples of Group 1; (**b**) Firmicutes, Proteobacteria, Actinobacteria, Bacteroidetes, and Fusobacteria were the five most abundant phyla in Group 1.

**Figure 5 diagnostics-11-01324-f005:**
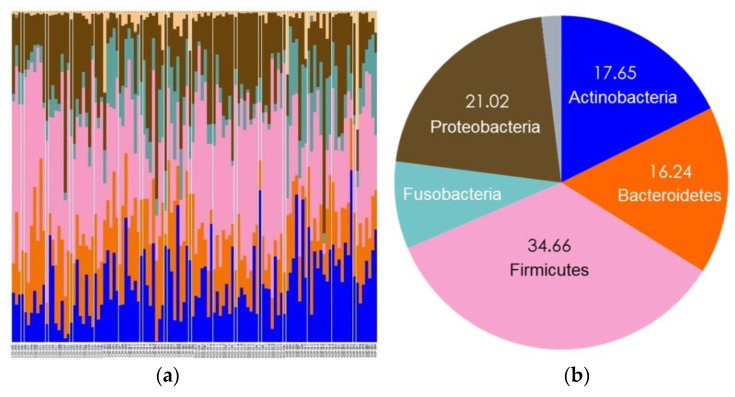
Taxonomic assignment at the phylum level in Group 2. (**a**) All taxa from the 120 saliva and plaque samples of Group 2; (**b**) Firmicutes, Proteobacteria, Actinobacteria, Bacteroidetes, and Fusobacteria were the five most abundant phyla in Group 2.

**Figure 6 diagnostics-11-01324-f006:**
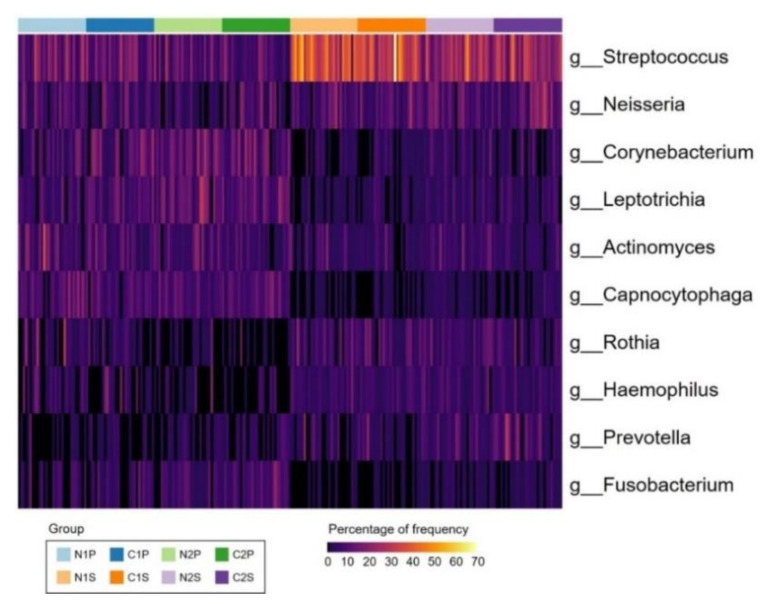
Heatmap of the 10 major genera. Each column represents a sample and each row a genus. The color bars correspond to the groups. The frequency of the genera in each sample was expressed as a percentage in a range of 0–70%. Higher proportion of yellow indicates more of those genera in the samples. N1P, plaque sample of the non-caries group of Group 1; N1S, saliva sample of the non-caries group of Group 1; N2P, plaque sample of the non-caries group of Group 2; N2S, saliva sample of the non-caries group of Group 2; C1P, plaque sample of the caries group of Group 1; C1S, saliva sample of the caries group of Group 1; C2P, plaque sample of the caries group of Group 2; C2S, saliva sample of the caries group of Group 2; g, genus.

**Figure 7 diagnostics-11-01324-f007:**
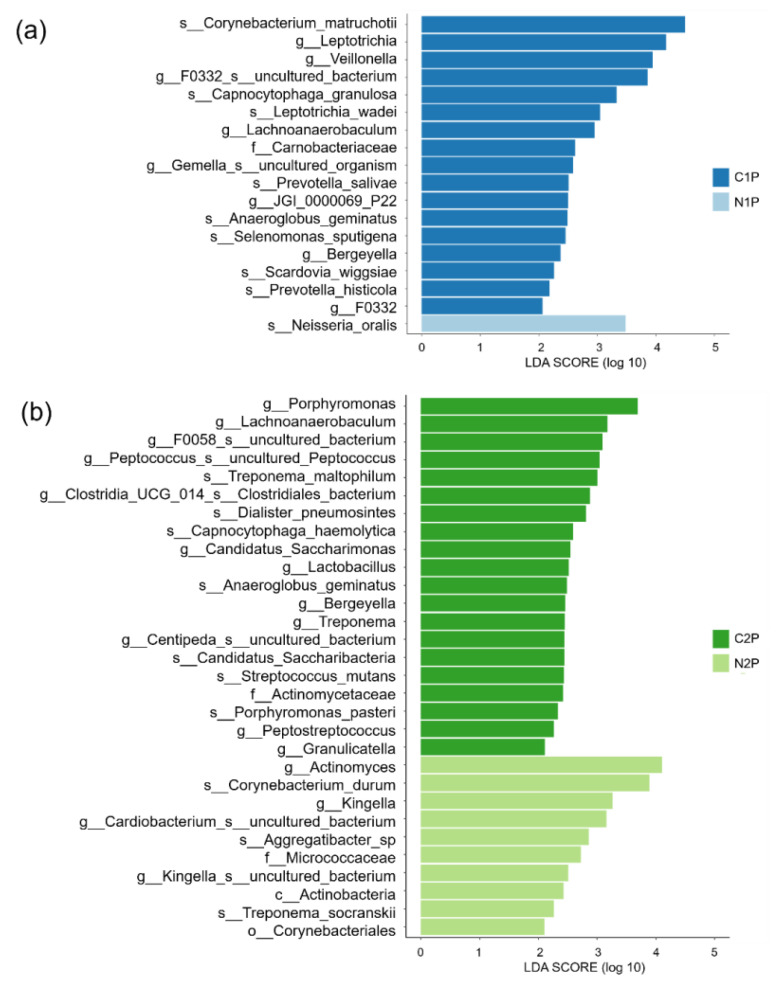
Histogram of differentially abundant taxa from the class to species level in the plaque samples of the non-caries and caries groups of Group 1 and Group 2 using linear discriminant analysis of effect size. (**a**) Taxa abundant in the plaque samples of the non-caries and caries groups of Group 1; (**b**) Taxa abundant in the plaque samples of the non-caries and caries groups of Group 2. The threshold of the logarithmic LDA score for discriminative features was set at 2.0; *p*-values <0.05 were considered statistically significant. LDA, linear discriminant analysis linear discriminant analysis; c, Class; o, Order; f, Family, g, Genus; s, Species; N1P, plaque sample of the non-caries group of Group 1; N2P, plaque sample of the non-caries group of Group 2; C1P, plaque sample of the caries group of Group 1; C2P, plaque sample of the caries group of Group 2.

**Figure 8 diagnostics-11-01324-f008:**
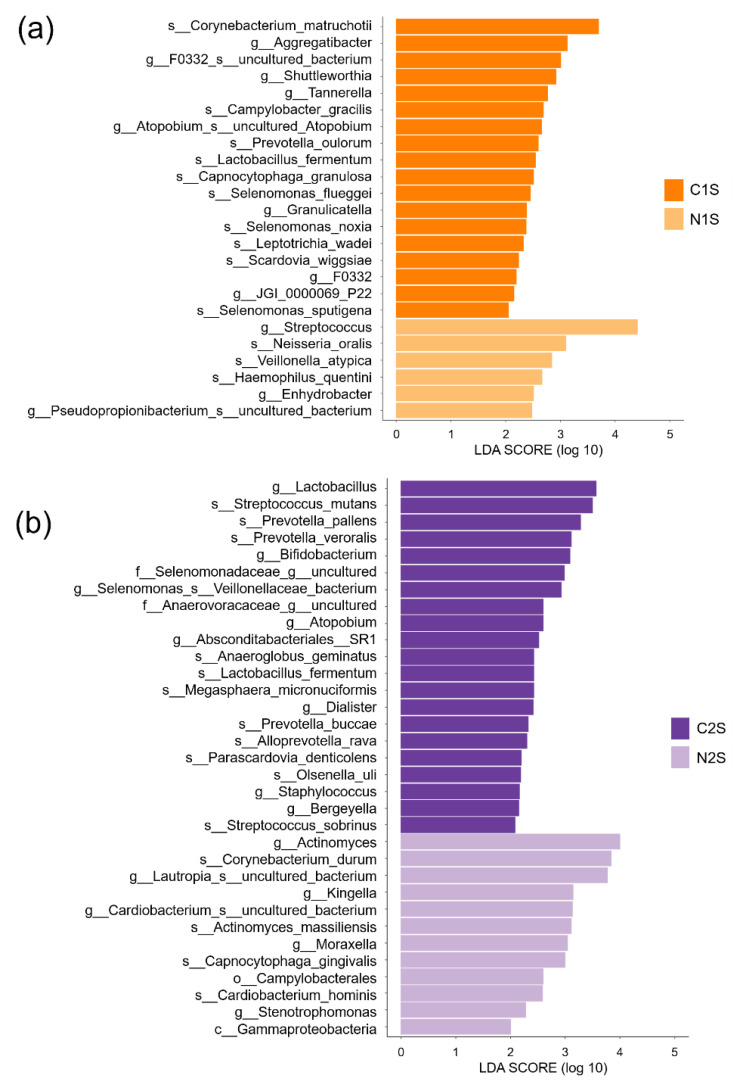
Histogram of the differentially abundant taxa from the class to species level in the saliva samples of the non-caries and caries groups of Group 1 and Group 2 using linear discriminant analysis of effect size. (**a**) Taxa abundant in the saliva samples of the non-caries and caries groups of Group 1; (**b**) Taxa abundant in the saliva samples of the non-caries and caries groups of Group 2. The threshold of the logarithmic LDA score for discriminative features was set at 2.0; *p*-values <0.05 were considered statistically significant. LDA, linear discriminant analysis; c, Class; o, Order; f, Family, g, Genus; s, Species; N1S, saliva sample of the non-caries group of Group 1; N2S, saliva sample of the non-caries group of Group 2; C1S, saliva sample of the caries group of Group 1; C2S, saliva sample of the caries group of Group 2.

**Table 1 diagnostics-11-01324-t001:** Distribution of the enrolled children by gender and age.

Groups		Gender	n	Mean Age (SD)	*p*-Value
Non-caries	(N1) ^1^	Male	10	4.3 (0.5)	0.731
		Female	20	4.4 (0.7)	
Caries	(C1) ^1^	Male	15	4.6 (0.7)	0.603
		Female	15	4.5 (0.7)	
Non-caries	(N2) ^2^	Male	14	8.1 (1.3)	0.560
		Female	16	7.9 (1.4)	
Caries	(C2) ^2^	Male	21	7.9 (1.3)	0.733
		Female	9	7.9 (1.4)	
Total			120	6.2 (2.0)	

^1^ Group 1 included children under 6 years of age; ^2^ Group 2 included children aged from 6 to 12 years. Significant differences were determined using an independent *t*-test at *p* < 0.05.

**Table 2 diagnostics-11-01324-t002:** Decayed teeth and International Caries Detection and Assessment System scores in the non-caries and caries groups.

Groups (n)	d + D ^1^	ICDAS ^2^ Score
	Mean (SD)	Mean (SD)
Non-caries (60)	0.00 (0.00)	0.30 (0.71)
Caries (60)	5.73 (3.51)	4.12 (0.78)
*p*-value	<0.001 *	<0.001 *

^1^ Decayed primary (d) and permanent (D) teeth based on the World Health Organization criteria; ^2^ International Caries Detection and Assessment System; * Significant differences were determined using an independent *t*-test at *p* < 0.05.

**Table 3 diagnostics-11-01324-t003:** Best performance with feature combination, accuracy, balanced accuracy, precision, sensitivity, and specificity in each group using random forest machine learning model.

Group	Feature Combination	Accuracy	Balanced Accuracy	Precision	Sensitivity	Specificity
N1P vs. C1P	*V + Lep*	0.78 (0.14)	0.81 (0.13)	0.80 (0.19)	0.80 (0.15)	0.81 (0.16)
N1S vs. C1S	*Sw + Cgra + Str + Cm + Gra + Agg + No + Lw*	0.70 (0.18)	0.71 (0.19)	0.67 (0.18)	0.74 (0.20)	0.69 (0.20)
N2P vs. C2P	*Cun + Gra + Lac + Por + CS + cbac + Cd + Kun*	0.73 (0.10)	0.75 (0.10)	0.73 (0.17)	0.74 (0.08)	0.75 (0.13)
N2S vs. C2S	*Sta + Act + Cun + B + Ato + Cgin*	0.83 (0.13)	0.84 (0.13)	0.87 (0.12)	0.82 (0.14)	0.86 (0.15)

*V, Veillonella; Lep, Leptotrichia; Sw, Scardovia wiggsiae; Cgra, Campylobacter gracilis; Str, Streptococcus; Cm, Corynebacterium matruchotii; Gra, Granulicatella; Agg, Aggregatibacter; No, Neisseria oralis; Lw, Leptotrichia wadei; Cun, Cardiobacterium uncultured bacterium; Lac, Lachnoanaerobaculum; Por, Porphyromonas; CS, Candidatus Saccharimonas; cbac, Clostridiales bacterium; Cd, Corynebacterium durum; Kun, Kingella uncultured bacterium; Sta, Staphylococcus; Act, Actinomyces; B, Bergeyella; Ato, Atopobium; Cgin, Capnocytophaga gingivalis.* N1P, plaque sample of the non-caries group of Group 1; N1S, saliva sample of the non-caries group of Group 1; N2P, plaque sample of the non-caries group of Group 2; N2S, saliva sample of the non-caries group of Group 2; C1P, plaque sample of the caries group of Group 1; C1S, saliva sample of the caries group of Group 1; C2P, plaque sample of the caries group of Group 2; C2S, saliva sample of the caries group of Group 2.

## Data Availability

The data presented in this study are available on request from the corresponding authors. The data are not publicly available due to ethical reasons.

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
