# Peer review of "Microbiome of Saliva and Plaque in Children According to Age and Dental Caries Experience"

_diagnostics, 2021, doi:10.3390/diagnostics11081324_

Round 1

Reviewer 1 Report

Many methodological biase exist, however it is an interesting article

(The Authors must see my remarks)

Author Response

Jul-20-2021

“Diagnostic Infectious Disease and Microbiology” of Diagnostics

I am pleased to submit this revised manuscript entitled “Microbiome of Saliva and Plaque in Children According to Age and Dental Caries Experience.” I really appreciate your review and comments and I can revise this manuscript based on your insight comments and recommendations.

I described all the details about your review comments, and I highlighted them in red colored text in this revision letter.

Thank you for your consideration. I look forward to hearing from you.

Sincerely,

Taesung Jeong, DDS, PhD

Department of Pediatric Dentistry, College of Dentistry, Pusan National University, Yangsan, Republic of Korea

Telephone: +82-55-360-5181

Fax: +82-55-360-5174

Response to Reviewer 1 Comments

Point 1: Please clarify the type of the article, eg. Research?

Response 1: We clarified the type of this article, “Research Article”.

Revised text: Please see title section (line 1, page 1).

Point 2: This Section must contain: Background/Aim - Materials and Methods - Results - Conclusion(s)...

Response 2: Thank you for your kind comment. We revised the abstract as journal’s recommendations.

Revised text: Please see abstract section (line 18-32, page 1).

Point 3: Avoid using parenthesis...

Response 3: We revised the abstract.

Revised text: Please see abstract section (line 18-32, page 1).

Point 4: Correct....

Response 4: Thank you for your comment.

Point 5: Replace by ''genders''...

Response 5: We changed sexes to genders.

Point 6: Moreover, cancer, R.arthritis, allergies, COPD, etc.....

Response 6: We revised and added more recent references.

Revised text: Please see instruction section (line 55-57, page 2).

Point 7: How did the Authors determine the study sample? Reference(s)? Protocol? Inclusion/Exclusion criteria? Selection/Recall biases? Confounders control?

Response 7: This study was designed with reference to the previous studies (Ref #13, 27, 59) and the study samples were calculated based on the methodologically similar studies (Ref #34,35). All participants who met the inclusion criteria completed all the procedures at a single visit.

All patients who visit the department of pediatric dentistry in Pusan National University Dental Hospital for the first time have to fill a simple questionnaire which is being used as basic data for the patients. The questionnaire consists of a few questions about the frequency of tooth brushing, whether to use fluoride toothpaste, and the experience of topical fluoride application, etc. Patients’ medical and/or dental histories can be checked and updated every appointment. Nonetheless, we feel deeply sorry that we could not fully consider the confounders in this study. We discussed about this point as limitations of our study in discussion section.

Revised text: Please see materials and methods section (line 89-100, page 2-3) and discussion section (line 499-502, page 14).

Point 8: Reference(s)?

Response 8: We inserted references.

Revised text: Please see materials and methods section (line 89, page 2).

Point 9: The reason(s)?

Response 9: Thank you for your comment. The study protocols were planned with reference to the previous studies (Ref #13, 27, 59). All clinical examinations and sample collections were performed by a specialized pediatric dentist in accordance with those references. However, we would like to increase the reliability of data through repeated examinations or examinations with two or more examiners in the future study as your comment.

Point 10: Intraexaminer variability (K-index)?

Response 10: As mentioned above, all clinical examinations and sample collections were performed once at a patient’s single visit. An assistant helped to record the findings, and the protocols were usually completed within 15 min. We originally planned not to repeat the procedures because the participants of this study included very young children like children aged 3 or 4. We will surely consider the intra/interexaminer variability in the future study with your comments in our minds.

Point 11: What about the deciduous teeth?

Response 11: Honkala et al. reported that ICDAS recording seems to give appropriate information from the occurrence of caries lesions and its correlations between the primary and permanent teeth (DOI: 10.1155/2011/150424). Dashper et al. used ICDAS-II in their clinical assessment when they researched that the oral microbiome related to the onset and progression of ECC and they showed significant results with this method (DOI: 10.1038/s41598-019-56233-0). We applied same criteria in permanent and deciduous teeth both with reference to these studies.

Point 12: Reference(s)?

Response 12: We inserted a reference.

Revised text: Please see materials and methods section (line 116, page 3).

Point 13: References for such criteria?

Response 13: Jiang et al. researched the oral microbiome in the elderly with saliva and plaque samples both from each subject (DOI: 10.3389/fcimb.2018.00442). However, there were not enough studies to figure out the oral microbiome of Korean children. In some other studies, researchers used saliva or plaque sample only and the number of subjects were too small. It was the starting point from which we planned and designed this study.

Point 14: Reference(s)?

Response 14: We inserted a reference.

Revised text: Please see materials and methods section (line 181, page ).

Point 15: Replace by ''gender''....

Response 15: We changed sex to gender.

Point 16: Was the distribution normal?

Response 16: We assured that the distribution was normal with the Shapiro-Wilk test.

Revised text: Please see materials and methods section (line 188, page 4).

Point 17: References are missing.....

Response 17: We added a reference.

Revised text: Please see materials and methods section (line 207, page 5).

Point 18: A linear regression model would be more useful... What about the possible confounders?

Response 18: Thank you for your comments. Linear regression analysis is mainly used when both the independent and dependent variables are continuous variables. Table 1 showed the distribution of the participants in this study, and it was confirmed that there was not significant in the age distribution by gender per group. Since gender is a categorical variable, and age is a continuous variable, we used t-test. We could not fully consider confounders such as individual’s oral hygiene habit, diet, fluoride application, etc. We discussed about this points in discussion section.

Point 19: Reference(s)?

Response 19: We added references.

Revised text: Please see discussion section (line 398, page 12).

Point 20: How do we know that?

Response 20: We revised in ‘Materials and Methods’ section.

Revised text: Please see materials and methods section (line 113-115, page 3)

Point 21: Reference(s)?

Response 21: We added a reference.

Revised text: Please see discussion section (line 440, page 13).

Point 22: More limitations exist, mainly methodological....

Response 22: Thank you for your kind comments. We discussed more limitations.

Revised text: Please see discussion section (line 499-502, page 14).

Reviewer 2 Report

This is a very interesting article that explores the differences in the microbiome of caries and non-caries children. Although the authors propose a compelling topic, several issues must be addressed before publication.

Abstract

The abstract does not provide a background to the issue.

 Introduction

Line 44: “ Although the number of microbes in the oral cavity is small as compared to the other organs in humans” – Which other organs are you referring to? I would disagree. Please add more details or justify this statement.

Age can have a major impact on various processes in the oral cavity, both physiological and pathological. The authors should also specify this, I suggest citing: Popa C, Filioreanu AM, Stelea C, Maftei GA, Popescu E. Prevalence of oral lesions modulated by patient's age: the young versus the elderly. Romanian Journal of Oral Rehabilitation. 2018 Jul 1;10(3):50-6.

What is the hypothesis of this study?

The introduction lacks focus, please reformulate so it is more cohesive.

Materials and methods:

  • Please specify whether consent was obtained from the parents?
  • When was saliva sampled?
  • What did the “E-zen Gargle, JN Pharm, Korea” solution contain? Did it interfere with saliva flow stimulation, composition?
  • Was power analysis performed to verify sample size?

Discussion:

  • Line 328: “Recently, laser fluorescence or quantitative optical fluorescence has also 328 been used.” – reference missing
  • The authors should discuss other possible implications of saliva composition besides microbiological assessment. I suggest citing: Solomon SM, Bataiosu M, Popescu DM, Rauten AM, Gheorghe DN, Petrescu RA, Maftei GA, Maglaviceanu CF. Biochemical Assessment of Salivary Parameters in Young Patients with Dental Lesions. Revista de Chimie. 2019 Nov 1;70(11):4095-7.
  • Plaque and bacterial composition are highly influenced by diet, the authors should discuss this aspect.
  • Also, bacterial composition in caries prone vs periodontal disease-prone individuals is highly different and can be influenced by systemic conditions and diseases, the authors should mention this aspect and I suggest citing: Martu, M.A., Maftei, G.A., Luchian, I., Popa, C., Filioreanu, A.M., Tatarciuc, D., Nichitean, G., Hurjui, L.L. and Foia, L.G. Wound healing of periodontal and oral tissues: part II - Patho-phisiological conditions and metabolic diseases.  Rom J of Oral Rehab.2020, 12(3).

Author Response

Jul-20-2021

“Diagnostic Infectious Disease and Microbiology” of Diagnostics

I am pleased to submit this revised manuscript entitled “Microbiome of Saliva and Plaque in Children According to Age and Dental Caries Experience.” I really appreciate your review and comments and I can revise this manuscript based on your insight comments and recommendations.

I described all the details about your review comments, and I highlighted them in red colored text in this revision letter.

Thank you for your consideration. I look forward to hearing from you.

Sincerely,

Taesung Jeong, DDS, PhD

Department of Pediatric Dentistry, College of Dentistry, Pusan National University, Yangsan, Republic of Korea

Telephone: +82-55-360-5181

Fax: +82-55-360-5174

Response to Reviewer 2 Comments

Point 1: The abstract does not provide a background to the issue.

Response 1: Thank you for your kind comment. We revised the abstract as journal’s recommendations.

Revised text: Please see abstract section (line 18-32, page 1).

Point 2: Line 44: “Although the number of microbes in the oral cavity is small as compared to the other organs in humans” – Which other organs are you referring to? I would disagree. Please add more details or justify this statement.

Response 2: We wanted to communicate that the gut has the greatest number of microbes in a human body. We revised to clarify this point and cited more references.

Revised text: Please see introduction section (line 51-53, page 2).

Point 3: Age can have a major impact on various processes in the oral cavity, both physiological and pathological. The authors should also specify this, I suggest citing: Popa C, Filioreanu AM, Stelea C, Maftei GA, Popescu E. Prevalence of oral lesions modulated by patient's age: the young versus the elderly. Romanian Journal of Oral Rehabilitation. 2018 Jul 1;10(3):50-6.

Response 3: Thank you for your insightful comments. We discussed this point in discussion section.

Revised text: Please see discussion section (line 479-483, page 13).

Point 4: What is the hypothesis of this study?

Response 4: This study was to identify the microbiome in the oral cavity in relation to dental caries. It started with the assumption that the oral microbiome of children at the risk for dental caries is different from that of children without dental caries. Numerous studies have shown that Streptococcus mutans is the primary pathogen for dental caries. However, since the microbial community in the oral cavity is different by ethnicity or age, this study was planned to identify major biomarkers other than Streptococcus mutans for causing dental caries in Korean children. In briefly, the null hypothesis was that there was no microbial differences between non-caries and caries group in children aged below 12 years.

Point 5: The introduction lacks focus, please reformulate so it is more cohesive.

Response 5: Thank you for your insightful comment. We revised the introduction.

Revised text: Please see introduction section (line 36-83, page 1-2).

Point 6: Please specify whether consent was obtained from the parents?

Response 6: We revised the manuscript in Materials and Methods section as your comments.

Revised text: Please see materials and methods section (line 106-108, page 3).

Point 7: When was saliva sampled?

Response 7: Clinical examinations and sample collections were performed on the same day. Saliva and plaque samples were collected in the morning from subjects who met the inclusion criteria. We revised the manuscript in Materials and Methods section as your comments.

Revised text: Please see materials and methods section (line 110-118, page 3).

Point 8: What did the “E-zen Gargle, JN Pharm, Korea” solution contain? Did it interfere with saliva flow stimulation, composition?

Response 8:

The ingredients of E-zen gargle are:

Cetylpyridinium chloride, Xylitol, Glucosyl Stevioside, Sodium Saccharin, Disodium EDTA, Glycerin, Blue No.1, Sodium citrate, Ethanol, Methylparaben, Polyoxyethylene Hydrogenated Castor Oil, Menthol, Mentha Oil, Purified water.

We used this commercially available mouthwash solution in Korea and the solution was used in other studies (Ref #39). Zaura et al. (DOI: 10.1111/prd.12359) and Vogtmann et al. (DOI: 10.1158/1055-9965.EPI-18-0312) showed higher alpha and beta diversity with samples using a commercially available mouthwash product than samples collected with only saliva, and the composition of oral microbiome was not significantly different. We cited a previous study used the same gargle solution.

Revised text: Please see materials and methods section (line 115-116, page 3).

Point 9: Was power analysis performed to verify sample size?

Response 9: This study was designed with reference to the previous studies (Ref #13, 27, 59) and the study samples were calculated based on the methodologically similar studies (Ref #34,35).

Revised text: Please see materials and methods section (line 89-93, page 2).

Point 10: Line 328: “Recently, laser fluorescence or quantitative optical fluorescence has also 328 been used.” – reference missing

Response 10: We marked references.

Revised text: Please see discussion section (line 398, page 12).

Point 11: The authors should discuss other possible implications of saliva composition besides microbiological assessment. I suggest citing: Solomon SM, Bataiosu M, Popescu DM, Rauten AM, Gheorghe DN, Petrescu RA, Maftei GA, Maglaviceanu CF. Biochemical Assessment of Salivary Parameters in Young Patients with Dental Lesions. Revista de Chimie. 2019 Nov 1;70(11):4095-7.

Response 11: We appreciate your reviews and comments. We discussed more with this point in the discussion section.

Revised text: Please see discussion section (line 485-488, page 13).

Point 12: Plaque and bacterial composition are highly influenced by diet, the authors should discuss this aspect. Also, bacterial composition in caries prone vs periodontal disease-prone individuals is highly different and can be influenced by systemic conditions and diseases, the authors should mention this aspect and I suggest citing: Martu, M.A., Maftei, G.A., Luchian, I., Popa, C., Filioreanu, A.M., Tatarciuc, D., Nichitean, G., Hurjui, L.L. and Foia, L.G. Wound healing of periodontal and oral tissues: part II - Patho-phisiological conditions and metabolic diseases.  Rom J of Oral Rehab.2020, 12(3).

Response 12: Thank you very much for your in-depth comments. We more discussed with these points in the discussion section.

Revised text: Please see discussion section (line 481-490, page 13 and line 499-502, page 14).

Reviewer 3 Report

Dear author,

This is an interesting article however some remarks should be taken into account

Introduction

  1. Ref 9 Please add more recent references (PMID: 31600905, PMID: 31591341, PMID: 30213346, https://doi.org/10.3390/ijerph18137194...)

Material and methods

  1. This section should be revised. Please precise the inclusion and exclusion criteria.
  2. Line 70 Add references
  3. Concerning group 1, what was the limit of age because young under 6 years with complete primary dentition is not enough precise.
  4. L90-91 Add reference
  5. L91-96 Precise how the sampling was done
  6. L97-100 Describe the methodology used more precisely: volume of saliva used, the quantity of plaque, elution volume….
  7. Precise how the number of children to include was calculated?
  8.  

Results

  1. Precise the min and max age for each group
  2. Table 2 It would be interesting to have these results for group 1 and group 2
  3. Figure 3 Represent the data of groups 1 and 2 with different colors
  4. Figure 7. “Histogram of differentially abundant taxa from the class to species level in the plaque 241 Scheme 1. and Group 2 using linear discriminant analysis of effect size.” Revise or delete “and Group 2 using linear discriminant analysis of effect size.” Because groups 1 and 2 are represented in this figure
  5.  

Discussion

  1. Line 371-375 Please compare your results with the ones observed in the interdental microbiota of adolescents or adults (doi:10.3390/microorganisms7090319; https://doi.org/10.1371/journal.pone.0185804)
  2. Please add as limitations that you didn’t take into account the social status or the habits of oral hygiene or nutrition that represent important risk factors of carious lesions.

Author Response

Jul-20-2021

“Diagnostic Infectious Disease and Microbiology” of Diagnostics

I am pleased to submit this revised manuscript entitled “Microbiome of Saliva and Plaque in Children According to Age and Dental Caries Experience.” I really appreciate your review and comments and I can revise this manuscript based on your insight comments and recommendations.

I described all the details about your review comments, and I highlighted them in red colored text in this revision letter.

Thank you for your consideration. I look forward to hearing from you.

Sincerely,

Taesung Jeong, DDS, PhD

Department of Pediatric Dentistry, College of Dentistry, Pusan National University, Yangsan, Republic of Korea

Telephone: +82-55-360-5181

Fax: +82-55-360-5174

Response to Reviewer 3 Comments

Point 1: Ref 9 Please add more recent references (PMID: 31600905, PMID: 31591341, PMID: 30213346, https://doi.org/10.3390/ijerph18137194...)

Response 1: Thank you for your review and comment. We revised the manuscript and added more recent references.

Revised text: Please see instruction section (line 53-57, page 2).

Point 2: This section should be revised. Please precise the inclusion and exclusion criteria.

Response 2: We revised the manuscript in Materials and Methods section as your comments.

Revised text: Please see materials and methods section (line 93-100, page 2-3).

Point 3: Line 70 Add references

Response 3: We added references.

Revised text: Please see materials and methods section (line 89, page 2).

Point 4: Concerning group 1, what was the limit of age because young under 6 years with complete primary dentition is not enough precise.

Response 4: We revised the manuscript in Materials and Methods section as your comments.

Revised text: Please see materials and methods section (line 93-97, page 2).

Point 5: L90-91 Add reference

Response 5: Jiang et al. researched the oral microbiome in the elderly with saliva and plaque samples both from each subject (DOI: 10.3389/fcimb.2018.00442). However, there were not enough studies to figure out the oral microbiome of Korean children. In some other studies, researchers used saliva or plaque sample only and the number of subjects were too small. It was the starting point from which we planned and designed this study.

Point 6: L91-96 Precise how the sampling was done

Response 6: Clinical examinations and sample collections were performed on the same day. Saliva and plaque samples were collected in the morning from subjects who met the inclusion criteria. We revised the manuscript in Materials and Methods section as your comments.

Revised text: Please see materials and methods section (line 110-118, page 3).

Point 7: L97-100 Describe the methodology used more precisely: volume of saliva used, the quantity of plaque, elution volume….

Response 7: We described the methodology more precisely as your comments.

Revised text: Please see materials and methods section (line 135-161, page 3-4).

Point 8: Precise how the number of children to include was calculated?

Response 8: This study was designed with reference to the previous studies (Ref #13, 27, 59) and the study samples were calculated based on the methodologically similar studies (Ref #34, 35).

Revised text: Please see materials and methods section (line 89-93, page 2).

Point 9: Precise the min and max age for each group

Response 9: We described about this point in results section

Revised text: Please see results section (line 226-228, page 5).

Point 10: Table 2 It would be interesting to have these results for group 1 and group 2

Response 10: Thank you for your kind comment. We have made the table for the results for group 1 and group 2 as well, but we thought that the current Table 2 could show the differences between groups more clearly.

Group (n)

d + D 1

ICDAS 2

Group (n)

d + D 1

ICDAS 2

Mean (SD)

Mean (SD)

Mean (SD)

Mean (SD)

N1 3 (30)

0.00 (0.00)

0.20 (0.60)

N2 5 (30)

0.00 (0.00)

0.40 (0.80)

C1 4 (30)

6.23 (3.59)

4.08 (0.62)

C2 6 (30)

5.23 (3.35)

4.15 (0.91)

p-value

< 0.001*

< 0.001*

p-value

< 0.001*

< 0.001*

1 Decayed primary (d) and permanent (D) teeth based on the World Health Organization criteria

2 International Caries Detection and Assessment System

3 N1: a group consisting of subjects under 6 years of age without dental caries

4 C1: a group consisting of subjects under 6 years of age with dental caries

5 N2: a group consisting of subjects aged from 6 to 12 years old without dental caries

6 C2: a group consisting of subjects aged from 6 to 12 years old with dental caries

* Significant differences were determined using an independent t-test at p < 0.05.

Point 11: Figure 3 Represent the data of groups 1 and 2 with different colors

Response 11: We really thank you for your suggestion. We represented the data of groups 1 and 2 with different colors.

Revised text: Please see results section (line 255-257, page 7).

Point 12: Figure 7. “Histogram of differentially abundant taxa from the class to species level in the plaque 241 Scheme 1. and Group 2 using linear discriminant analysis of effect size.” Revise or delete “and Group 2 using linear discriminant analysis of effect size.” Because groups 1 and 2 are represented in this figure

Response 12: Thank you for your kind comments. We checked the summited our original manuscript, and we confirmed that we wrote as “Histogram of differentially abundant taxa from the class to species level in the plaque samples of the non-caries and caries groups of Group 1 and Group 2”. We think it must have been a system error, so please let us know if it still looks the same.

Point 13: Line 371-375 Please compare your results with the ones observed in the interdental microbiota of adolescents or adults (doi:10.3390/microorganisms7090319; https://doi.org/10.1371/journal.pone.0185804)

Response 13: We really thank you for your insightful comments and suggestions. We revised and cited references as your comments.

Revised text: Please see discussion section (line 443-447, page 13).

Point 14: Please add as limitations that you didn’t take into account the social status or the habits of oral hygiene or nutrition that represent important risk factors of carious lesions.

Response 14: Thank you for your kind comments. We discussed more limitations in discussion section.

Revised text: Please see discussion section (line 499-502, page 14).

Round 2

Reviewer 2 Report

The manuscript has been significantly improved

Reviewer 3 Report

Thank you for considering my comments